# Timeliness of routine childhood vaccination among 12–35 months old children in The Gambia: Analysis of national immunisation survey data, 2019–2020

Oghenebrume Wariri[1,2,3]*, Chigozie Edson Utazi[4,5], Uduak Okomo[1,6], Malick Sogur[7], Kris A. Murray[8], Chris Grundy[2], Sidat Fofanna[7], Beate Kampmann[1,9]

1 Vaccines and Immunity Theme, MRC Unit The Gambia at London School of Hygiene and Tropical Medicine, Fajara, The Gambia, 2 Department of Infectious Disease Epidemiology, London School of Hygiene and Tropical Medicine, London, United Kingdom, 3 Vaccine Centre, London School of Hygiene and Tropical Medicine, London, United Kingdom, 4 WorldPop, School of Geography and Environmental Science, University of Southampton, Southampton, United Kingdom, 5 Southampton Statistical Sciences Research Institute, University of Southampton, Southampton, United Kingdom, 6 MARCH Centre, London School of Hygiene and Tropical Medicine, London, United Kingdom, 7 Expanded Programme on Immunization, Ministry of Health and Social Welfare, Banjul, The Gambia, 8 Centre on Climate Change and Planetary Health, MRC Unit The Gambia at The London School of Hygiene and Tropical Medicine, Fajara, The Gambia, 9 Centre for Global Health, Charite Universitatsmedizin Berlin, Berlin, Germany

* Oghenebrume.Wariri@lshtm.ac.uk

**Data Availability Statement:** The data underlying the results presented in this study were collected as part of the Gambia Demographic and Health

## Abstract

The Gambia's routine childhood vaccination programme is highly successful, however, many vaccinations are delayed, with potential implications for disease outbreaks. We adopted a multi-dimensional approach to determine the timeliness of vaccination (i.e., timely, early, delayed, and untimely interval vaccination). We utilised data for 3,248 children from The Gambia 2019–2020 Demographic and Health Survey. Nine tracer vaccines administered at birth and at two, three, four, and nine months of life were included. Timeliness was defined according to the recommended national vaccination windows and reported as both categorical and continuous variables. Routine coverage was high (above 90%), but also a high rate of untimely vaccination. First-dose pentavalent vaccine (PENTA1) and oral polio vaccine (OPV1) had the highest timely coverage that ranged from 71.8% (95% CI = 68.7–74.8%) to 74.4% (95% CI = 71.7–77.1%). Delayed vaccination was the commonest dimension of untimely vaccination and ranged from 17.5% (95% CI = 14.5–20.4%) to 91.1% (95% CI = 88.9–93.4%), with median delays ranging from 11 days (IQR = 5, 19.5 days) to 28 days (IQR = 11, 57 days) across all vaccines. The birth-dose of Hepatitis B vaccine had the highest delay and this was more common in the 24–35 months age group (91.1% [95% CI = 88.9–93.4%], median delays = 17 days [IQR = 10, 28 days]) compared to the 12–23 months age-group (84.9% [95% CI = 81.9–87.9%], median delays = 16 days [IQR = 9, 26 days]). Early vaccination was the least common and ranged from 4.9% (95% CI = 3.2–6.7%) to 10.7% (95% CI = 8.3–13.1%) for all vaccines. The Gambia's childhood immunization system requires urgent implementation of effective strategies to reduce

Survey (DHS) conducted in 2019-2020. This data is publicly available for download from the DHS Program website at https://dhsprogram.com/data/available-datasets.cfm, however, it is important to note that prior permission from the DHS program must be obtained before accessing the data.

**Funding:** Funding This project is part of the EDCTP2 Programme supported by the European and Developing Countries Clinical Trials Partnership (grant number TMA2019CDF-2734 – TIMELY). OW was also supported by an Imperial College London Wellcome Trust Institutional Strategic Support Fund (ISSF) (grant no RSRO_P67869). CEU was supported by funding from the Bill & Melinda Gates Foundation (Investment ID INV-003287). The Vaccines and Immunity Theme (OW, UO, and BK) is jointly funded by the UK MRC and the UK Department for International Development (DFID) under the MRC/DFID Concordat agreement and is also part of the EDCTP2 Programme supported by the EU (MC UP_A900/1122, MC UP A900/115). The funders had no role in study design, data collection and analysis, decision to publish, or preparation of the manuscript.

**Competing interests:** The authors have declared that no competing interests exist.

untimely vaccination in order to optimize its quality, even though it already has impressive coverage rates.

## Introduction

The Gambian routine childhood vaccination programme is highly successful; for over a decade, it has consistently maintained routine childhood vaccination coverage rates of at least 90% for most routine childhood vaccines [1, 2]. The Gambia is therefore considered a model for the delivery and of coverage of routine childhood vaccines for many sub-Saharan African countries. The country achieved the 2020 Global Vaccine Action Plan (GVAP) coverage target a decade early [1], and is on track to reach the coverage target of the 2030 immunization agenda (IA2030) which aims to achieve at least 90% coverage for routine childhood vaccines [3]. Despite the celebrated success, there is growing evidence that many children are not receiving their vaccines within the recommended time frames [4–6]. This is particularly worrying as in parts of The Gambia, there has been a recent upsurge of measles, with a 6-fold increase in cases as of mid-2022 compared to 2020 figures despite a high coverage of both doses of measles-containing vaccine (MCV1 and MCV2) [7, 8]. Measles outbreaks are considered a sensitive marker of emerging herd immunity gaps [9] that might be created by untimely vaccination even in populations with otherwise high routine measles vaccination coverage rates [10].

Timely vaccination, operationally defined as vaccination received within the recommended age windows (i.e., valid doses) [11, 12], explores the quality dimension of immunization programs and is important for EPI programs like The Gambia that have already achieved high overage rates for most vaccines [13]. Deciding the appropriate window for childhood vaccination depends on several factors such as local disease epidemiology, presence of maternal antibodies, and the earliest age at which vaccines can be safely administered with maximum efficacy and the lowest risk of adverse effects [14]. Early vaccination, i.e., vaccines received before the earliest recommended window, may result in suboptimal immune response as maternal antibodies may inhibit vaccine response [14–18]. Conversely, delayed vaccination, i.e., vaccination received after the latest recommended window, prolongs the exposure of children to potentially life-threatening but vaccine-preventable diseases (VPDs) such as pertussis and measles [14, 19]. There are reports from other settings showing that measles outbreaks have occurred despite high vaccination coverage rates, suggesting a link to untimely vaccination [20]. Furthermore, delayed vaccination may have a domino effect on timeliness of other routine vaccines resulting in a child not completing their required vaccinations or receiving successive doses of a multi-series vaccine in an untimely manner (i.e., untimely interval vaccination) [21].

At the programmatic level, vaccinations given too early (before their earliest recommended window) or that are delayed (after the recommended windows) are key indicators for monitoring and evaluating the quality of an immunization program [22]. Untimely vaccination could be the only early warning signal that may alert the immunization system to potential problems with the delivery and uptake of vaccines. Timeliness of vaccination also has implications for the introduction of novel vaccines. For example, the World Health Organization (WHO) initially placed a strict age limit on the administration of rotavirus vaccine, stating it should not be initiated in infants aged 12 weeks or older to minimize the potential risk of intussusception, a rare form of bowel obstruction [23]. This policy restricted rotavirus vaccine introduction in many low-and middle-income countries (LMICs) where untimely vaccination was a main concern [24].

Studies exploring childhood vaccination timeliness in LMICs have gained momentum in the last decade [25]. In The Gambia, three studies, published in 2014 [6], 2015 [5], and 2016 [4] have so far assessed childhood vaccination timeliness. Nevertheless, many of the studies from LMICs, including the Gambian studies have key methodological issues that limit their utility and comparability. First, previous studies have primarily focused on delayed vaccination, with limited research into other crucial dimensions such as early vaccination and untimely interval vaccination for multi-series vaccines [25]. This one-dimensional approach provides inadequate data needed to gain a holistic understanding of untimely vaccination. Second, most of the previous studies operationalized vaccination timeliness as a categorical variable, mainly reporting the proportion of children with untimely vaccination [25]. While this approach appears pragmatic, it is simplistic, lumping together a wide window of untimely vaccinations and preventing a nuanced interpretation of the outcome. Populations with comparatively longer mean or median number of days children were vaccinated too early or delayed, outside the recommended windows, potentially have a higher likelihood of suboptimal immune response or risk of VPD outbreaks. Third, previous studies did not compare vaccination timeliness to official national routine vaccination coverage rates. Lastly, none of the Gambian studies used a nationally representative data; consequently, their findings give an incomplete picture of the true scale of untimely vaccination in The Gambia. This study, therefore, aims to bridge all the identified gaps by utilizing nationally-representative data to comprehensively investigate all dimensions of routine childhood vaccination timeliness and present categorical and continuous outcomes across two birth cohorts in The Gambia.

## Materials and methods

### Study setting and context

The Gambia is located in West Africa, with a population of about 2.5 million people and a birth cohort of 90,000 children who are added to the routine childhood immunization program yearly [26]. The national expanded programme on immunization (EPI) was launched in May 1979 and initially delivered six vaccines targeting tuberculosis (BCG vaccine), diphtheria, pertussis, tetanus (combined DTP vaccine), measles, polio, and yellow fever. The current childhood vaccination schedule include vaccines administered at birth and at two, three, four, nine, twelve and eighteen months of life (Table 1) [27]. We explored the timeliness of vaccination using tracer vaccines given in the first year of life, a period when the peaks and severity of VPDs are highest. The included vaccines are Bacilli Calmette Guerin (BCG) and the birth dose of Hepatitis B vaccine (HepB0) administered at birth; the first, second, and third doses of multi-series oral polio vaccine (OPV) and pentavalent vaccine (diphtheria, tetanus, pertussis, hepatitis B, and Haemophilus influenzae type b) given at two, three and four months of life; and the first dose of measles containing vaccine (MCV1), which is administered at nine months in The Gambia (Table 1).

### Data sources, study design and population

We analysed vaccination data from The Gambia Demographic and Health Survey (DHS), 2019–2020. The DHS is a nationally representative household survey that was designed by the global DHS program and implemented by the Gambia Bureau of Statistics (GBoS) [30]. The design, and implementation of the DHS is described in detail elsewhere [30]. In brief, The Gambia DHS 2019–2020 was performed using a two-stage cluster sampling design. In the first stage, the DHS selected a random sample of clusters with a probability proportional to their size within each sampling stratum from an already existing sample frame that was based on an updated version of the 2013 Gambia Population and Housing Census. In the second stage,

**Table 1. The Gambia routine childhood immunization schedule showing vaccines given during infancy and the accepted national vaccination window [27].**

| Vaccine | Vaccination window (Timely or age-appropriate vaccination) | Early vaccination | Delayed vaccination |
|---|---|---|---|
| Hepatitis B vaccine birth dose (HepB0)* | Birth | NA | > 24 hours of life [28] |
| Bacilli Calmette Guerin (BCG)* | | NA | > 7 days [29] |
| Oral Polio Vaccine (OPV0) | | | |
| Oral Polio Vaccine (OPV1)* | 2 Months (61–90 days) | <61 days | >90 days |
| Pentavalent vaccine (PENTA1)* | | <61 days | >90 days |
| Pneumococcal vaccine (PCV1) | | | |
| Rotavirus vaccine (Rota1) | | | |
| Oral Polio Vaccine (OPV2)* | 3 Months (91–120 days) | <91 days | >120 days |
| Pentavalent vaccine (PENTA2)* | | <91 days | >120 days |
| Pneumococcal vaccine (PCV2) | | | |
| Rotavirus vaccine (Rota2) | | | |
| Oral Polio Vaccine (OPV3)* | 4 Months (121–150 days) | <121 days | >150 days |
| Pentavalent vaccine (PENTA3)* | | <121 days | >150 days |
| Pneumococcal vaccine (PCV3) | | | |
| Inactivated Polio Vaccine (IPV) | | | |
| Measles and Rubella vaccine (MCV1)* | 9 Months (271–300 days) | <271 days | >300 days |
| Oral Polio Vaccine (OPV4) | | | |
| Yellow Fever vaccine | | | |
| OPV1 –OPV2; OPV2 –OPV3 interval* | 4–8 weeks (28–56 days) | <4 weeks or 28 days** | >8 weeks or 56 days** |
| PENTA1 –PENTA2; PENTA2 –PENTA3 interval* | 4–8 weeks (28–56 days) | <4 weeks or 28 days** | >8 weeks or 56 days** |

**Note**: *The tracer vaccines and vaccination intervals examined in this study. ** untimely interval represents the combination of both scenarios. Pentavalent vaccine = DPT-HepB-Hib.

households were systematically sampled from each cluster. The samples were stratified by urban and rural areas and sample weights were determined that must be applied to generate statistics that are representative at the national, urban and rural levels, and at the Local Government Area levels (i.e., first administrative area). The Gambia DHS 2019–2020 was conducted from 21 November 2019 to 30 March 2020 in 7025 selected households [30].

The Gambia DHS 2019–2020 collected childhood immunization data from 5,148 children aged 0–35 months who received specific vaccines at any time before the survey based on information from the child's health card or the mother's recall of vaccination. Overall, 93% of the included children (0–35 months) had vaccination cards, thus, accurate information on date of birth, vaccines received, and the dates of receipt (the variables needed to compute vaccination timeliness) were extracted directly from these cards. To ensure our timeliness analyses were comparable to the routine childhood coverage estimates routinely published in the DHS final reports [30], we generated timeliness output for the 3,248 children across two age groups: 12–23 months; and 24–35 months. A comprehensive breakdown detailing the number of children in the included age group and the availability of their birth and vaccination dates for the calculation of vaccination timeliness can be found in the supporting information (S1 Table).

## Measuring the dimensions of vaccination timeliness

At the individual level, we used the difference between vaccination dates and birth date to determine the age at vaccination (in days) for every vaccine. Using the nationally accepted childhood vaccination window in The Gambia (Table 1) [27], we converted the accepted age recommendations given in months and weeks to days. To ensure uniformity and

comparability, we considered a month to be equal to 30 days and a week was equal to 7 days. We considered each recommended age to begin at the first day of the window and end at the greatest number of days that could compose the given number of months or weeks (Table 1). For example, for vaccines that are recommended at two months of life, ***timely vaccination*** (or "on time") was any dose received between 61 days (the first day the child turned two months) and 90 days (the last day the child was two months). Any vaccination administered outside of the accepted window was considered "untimely" and include the following dimensions; early vaccination, delayed vaccination and untimely interval vaccination.

**Early vaccination.**    This was defined as vaccines received *before the earliest* nationally accepted valid ages or vaccination window (in days) for a specific vaccine in The Gambia (Table 1). Since BCG and HepB0 are recommended at birth, they can either be timely or delayed and cannot be administered too early unlike the other tracer vaccines.

**Delayed vaccination.**    This was defined as vaccines received *after the latest* nationally accepted valid ages or vaccination window (in days) for a specific vaccine in The Gambia (Table 1). The WHO recommends that infants receive HepB0 as soon as possible after birth, preferably within 24 hours [28], thus, delayed HepB0 was defined as doses received after 24 hours of life (i.e., 2 days and above). For BCG which is also recommended to be given "as soon as possible after birth", we instead used The Gambia Ministry of Health's definition of BCG received after 7 days (one week) as delayed [29].

**Untimely interval vaccination.**   In line with the WHO guideline [31], the recommended interval for subsequent doses of multi-series vaccine in The Gambia is 4–8 weeks (i.e., 28–56 days). Thus, we defined "untimely interval vaccination" as any subsequent dose of a multi-series vaccine received before or after the recommended window (i.e., interval $<28$ days or $>56$ days between doses).

## Data analysis

Following the DHS direct survey methodology, we computed routine vaccination coverage as the proportion of all eligible children (i.e., 12–23 months and 24–35 months) who were vaccinated. The denominator for computing routine vaccination coverage was all eligible children, within the specified age ranges (12–23 months and 24–35 months), with any form of vaccination evidence (i.e., either vaccination cards or mother's recall).We subsequently computed vaccination timeliness (timely, early, delayed and untimely interval vaccination) among those who were vaccinated. The denominator for computing vaccination timeliness (timely, early, delayed and untimely interval vaccination) was all eligible children whose date of birth was known, were within the specified age ranges, and whose vaccination dates were available from a vaccination card [12]. We report the proportion vaccinated (coverage, timely, and untimely vaccination) and 95% confidence interval (CI). We also computed vaccination timeliness as continuous variables and reported the median days (and interquartile range) outside the accepted window that children were vaccinated too early or too late (delayed). In all statistical analyses, we accounted for survey design and sample weights following standardized techniques [32], implemented using the *survey package* in R [33]. All analyses were performed in Rand figures were generated using the ggplot2 package [34].

## Ethics

This study was based on the analysis of the openly available The Gambia demographic and health survey 2019/2020 (GDHS 2019–2020). The ethical procedures for GDHS 2019–2020 were the responsibilities of the institutions that commissioned, funded or managed the surveys. The DHS states that it sought written informed consent from all participants before data

collection and the study did not include minors. We received formal approval from the DHS program to use GDHS 2019–2020 dataset. The Gambia Government and MRC Unit The Gambia at LSHTM Joint Ethics Committee granted ethical approval for secondary data analysis (Project ID/Ethics ref: 22786; Date: 16 January, 2021).

## Results

### Pattern of routine vaccination coverage compared to timely vaccination

The routine vaccination coverage was high (90% or more) across all tracer vaccines. However, many children were vaccinated outside the recommended national vaccination windows as shown in the cumulative routine vaccination coverage curve in Fig 1.

Fig 2 shows the comparison between routine coverage and timely vaccination coverage in The Gambia. Overall, the coverages of all the included childhood vaccines were above 90% in the two age groups, except the coverage of OPV3 which was 88% (95% CI = 85.7–90.2) in the 24–35 months age group.

The percentage of children with timely or "on time" vaccination was higher in the 12–23 months compared to the 24–35 months age group. Timely vaccination ranged from 15.1% (95% CI = 12.1–18.1%) to 74.4% (95% CI = 71.7–77.1%) in the 12–23 months age group compared to 8.9% (95% CI = 6.6–11.1%) to 71.9% (95% CI = 68.8–75.0%) in the 24–35 months age group (Fig 1). For specific childhood vaccines, timely vaccination was lowest for the vaccines scheduled to be administered at birth, especially HepB0 and the pattern was similar in the 12–23 and 24–35 age group. Timely vaccination was highest for the vaccines scheduled to be administered at two months of life (OPV1 and Penta1), which corresponds to the next contact with the vaccination system outside the birth period (Fig 2).

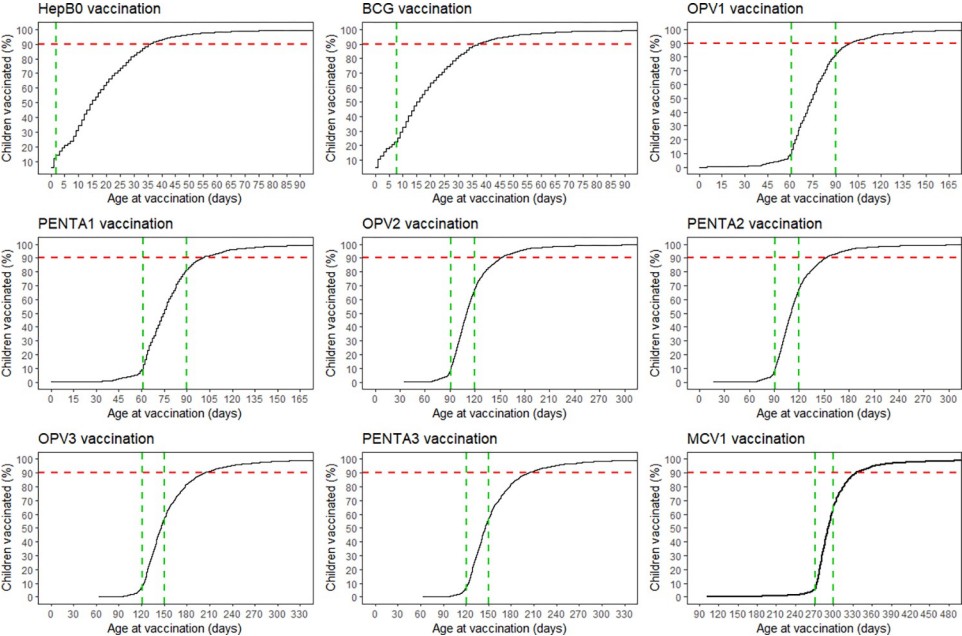

**Fig 1. Cumulative routine vaccination coverage curve of children 12–35 months in The Gambia.** *Note*: Green vertical dotted lines indicate the recommended national vaccination windows (≤24 hours for HepB0, 1–7 days for BCG, 61–90 days for OPV1 and PENTA1, 91–120 days for OPV2 and PENTA2, 121–150 days for OPV3 and PENTA3, and 271–300 days for MCV1). Red dotted line indicates WHO routine vaccination coverage target. Denominator is all eligible children with any evidence of vaccination, including vaccination cards and maternal recall.

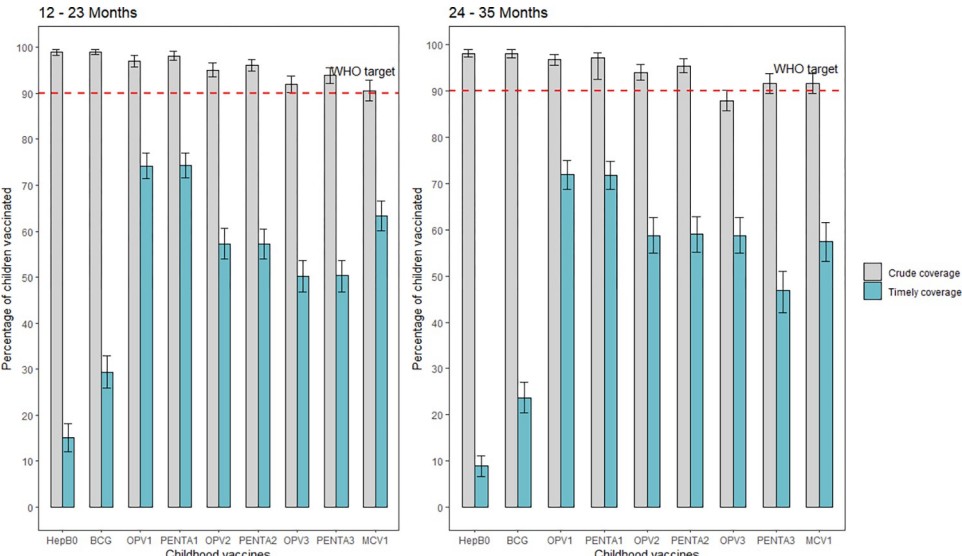

**Fig 2. The comparison between routine vaccination coverage and timely vaccination coverage in The Gambia.**
*Note*: Red horizontal dotted lines indicate the 2020 Global Vaccine Plan (GVAP) and the Immunization Agenda 2030 (IA2030) country-level crude vaccination coverage target. Error bar indicates 95% confidence interval (CI).

### Early vaccination (categorical and continuous outcomes)

The proportion of children who received their vaccinations too early was lower compared delayed vaccination. Overall, early vaccination ranged from 4.9% (95% CI = 3.2–6.7) to 10.7% (95% CI = 8.3–13.1) and the proportions were similar in the two age group (Fig 3A). The median number of days children were vaccinated too early, before the recommended window ranged from 3 days (IQR = 1, 11 days) to 14.5 days (IQR = 6.75, 26.25 days) and has a similar pattern in the two age groups (Fig 3B, supporting information [S2 Table]).

As per specific vaccines, the percentage of children with early vaccination was lowest for the vaccine that protect against measles infection, MCV1 (5.1% vs 4.9% in the 12–23 and 24–35 months age groups respectively). However, MCV1 had the highest median number of days that children were vaccinated too early (14.5 days, [IQR = 6.75, 26.25] and 9.0 days, [IQR = 4.0, 26.0] for 12–23 and 24–35 months age group respectively) compared to other vaccines (Fig 3B).OPV2 (in the 12–23 months) and OPV1 (in the 23–35 months) had the highest proportion of children with early vaccination representing 10.7% (95% CI = 8.3–13.1) and 10.6% (95% CI = 8.4–12.8), respectively (Fig 3A).

### Delayed vaccination (categorical and continuous outcomes)

Delayed vaccination ranged from 17.5% (95% CI = 14.5–20.4) to 91.1% (95% CI = 88.9–93.4), showing a similar pattern in the two age groups (Fig 4A). The median number of days children were vaccinated too late (i.e., delayed) ranged from 11 days (IQR = 5, 19.5 days) to 28 days (IQR = 11, 57 days) across all vaccines and had a similar pattern in the two age groups (Fig 4B, supporting information [S2 Table]). Overall, HepB0 had the highest proportion of delayed vaccination across the two age groups. Specifically, delayed HepB0 was higher in the 24–35 months age group (91.1% [95% CI = 88.9–93.4], median days delayed = 17 days [IQR = 10, 28 days]) compared to the 12–23 months age group (84.9% [95% CI = 81.9–87.9], median days delayed = 16 days [IQR = 9, 26 days]).

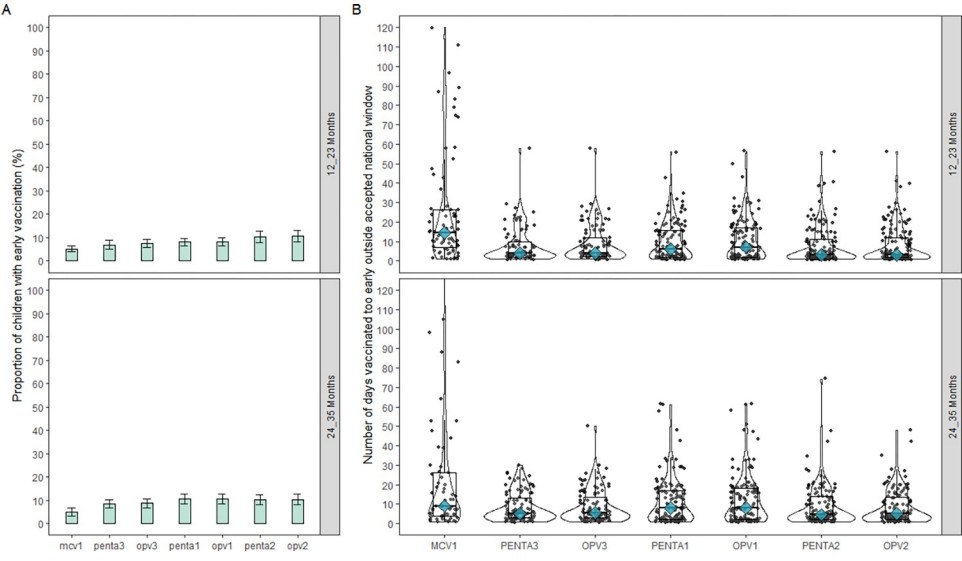

**Fig 3.** Early vaccination among children 12–23 months and 24–35 months in The Gambia (a) the proportion with early vaccination (categorical timeliness), (b) Number of days before the earliest accepted window that children were vaccinated too early (continuous timeliness). Note: Panel B is truncated at 120 days, thus, does not show the number of days outside the window for the outliers.

The percentage of children with delayed vaccination was lowest for the first dose of the multi-series vaccines (i.e., OPV1 and Penta1), with a gradual increase in delayed vaccination with subsequent doses in the series across the two age groups (Fig 4A). Similarly, the median number of days that children were vaccinated too late for OPV1 in the 24–35 months age group increased from 11 days (IQR = 4, 29 days) until it peaked at 28 days (IQR = 11, 57 days) for OPV3 (Fig 4B, supporting information [S2 Table]). The 12–23 months age group also

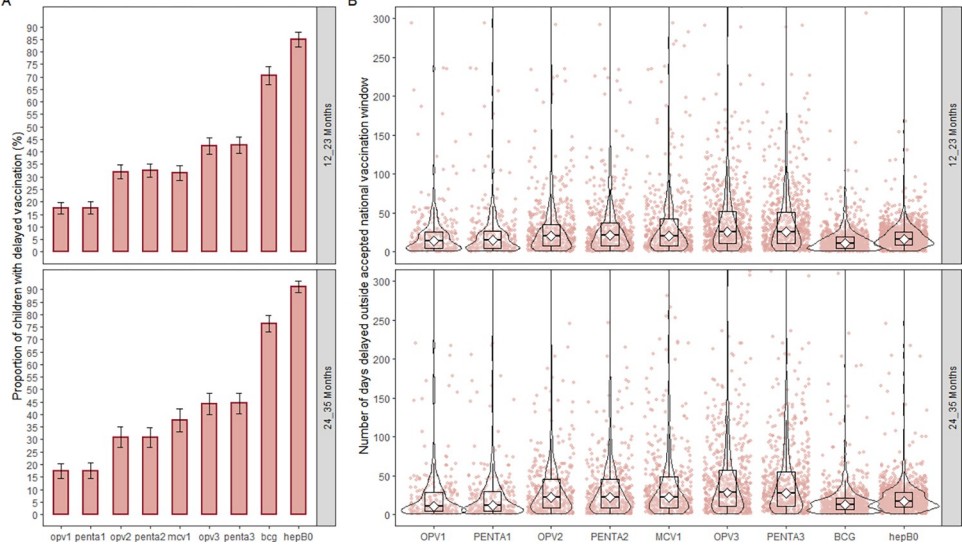

**Fig 4.** Delayed vaccination among children 12–23 months and 24–35 months in The Gambia (a) the proportion with delayed vaccination (categorical timeliness), (b) Number of days after the latest accepted window that children were vaccinated too late or delayed (continuous timeliness). Panel B is truncated at 300 days, thus, does not show the number of days outside the window for the outliers.

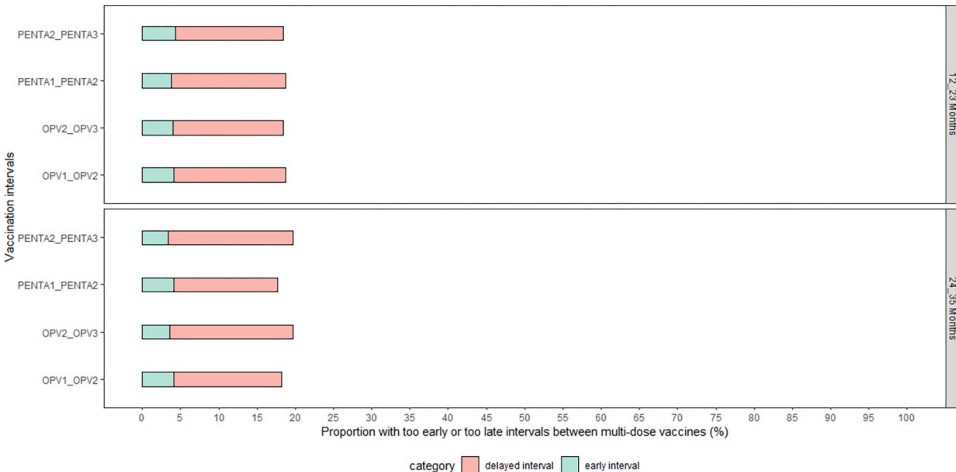

**Fig 5. The proportion of children with untimely interval vaccination for subsequent doses of multi-series vaccines in The Gambia. Note**: continuous timeliness (number of days outside the interval) was not computed for this dimension of vaccination timeliness because "untimely interval" include those who were vaccinated before and after the recommended interval for the multi-series vaccines.

followed a similar pattern of increasing median days children were vaccinated too late with subsequent doses of the multi-series vaccines.

## Untimely interval of vaccination between multi-series vaccines

Overall, less than 20% of the vaccinated children received their multi-series vaccines outside of the recommended window (i.e., a minimum interval of four weeks and maximum interval of eight weeks between subsequent doses of multi-series vaccines). This trend was observed consistently across both age cohorts and across all multi-series vaccines (Fig 5). Too early interval (i.e., being vaccinated less than 4 weeks or 28 days between subsequent doses of multi-series vaccines) was the least common, accounting for less than 5% for all multi-series vaccines (Fig 5).

## Discussion

To the best of our knowledge, this is the first study conducted in an LMIC context to simultaneously investigate all the dimensions of vaccination timeliness (timely, early, delayed, and untimely interval), and presents the outcomes as both categorical and continuous variables. This approach allows for a uniquely nuanced interpretation of the results unlike previous studies that often focused on one dimension or predominantly reported the outcomes as categorical variables. We also compared vaccination timeliness to official national survey-based routine vaccination coverage rates. Although overall coverage was high, a large number of children were vaccinated outside the recommended time-frames. Early vaccination was the least common dimension of untimely vaccination and also had a comparatively shorter median number of days children were vaccinated outside the window. Delayed vaccination was the most common dimension of untimely vaccination, with the highest proportion and longest median number of days children were vaccinated outside the recommended time-frames. Our findings do not align with prior research on vaccination timeliness, as the proportion of delayed vaccinations and the median delays in our study are generally lower compared to the largest study so far that included data of 217,706 children from 45 LMICs [35]. Our findings demonstrate that the Gambia EPI not only achieves high routine childhood vaccination

coverage rates but has also ensured that children receive their vaccinations within the recommended time-frames, as much as possible, in comparison to other LMICs.

In the last decade, The Gambian EPI program has further strengthened its commitment to leave no child behind and to reach 100% immunization coverage in the country. This commitment is supported by development partners such as Gavi, the Vaccine Alliance which continues to make huge investments towards ensuring that all children receive all their basic vaccinations within the recommended time-frames [36]. These commitments and investments might explain why vaccination was generally more timely in the younger age group (12–23 months) compared to children in the older age group. This improvement in timely vaccination is similar to the official national survey-based routine vaccination coverage estimates which shows that coverage was also higher among children in the younger age group [30].

For the multi-series vaccines, the proportion and median delays increased gradually and peaked with the third doses, reflecting a pattern similar to previous studies from Indonesia [37], the UK [38], and across LMICs context [35]. This trend is not surprising because the first doses of the multi-series vaccines are administered at two months of life in The Gambia which coincides with the first vaccination visit outside the birth period and may also be an opportunity to receive post-natal services, hence, the timely uptake. Nevertheless, it is worrisome because it reflects the inability of the program to consistently ensure timely vaccination or a lack of enthusiasm by caregivers for subsequent doses of multi-series vaccines. This situation may have a knock-on effect as many children may progress from untimely vaccination in subsequent doses to actual dropout from the system with increasing median delays with subsequent doses. Our results should, therefore, inform the development of retention strategies by the EPI for multi-series vaccines aimed at delivering doses in a timely manner.

## Implications for childhood vaccination planning and policy in The Gambia

The fact that HepB0 had the highest proportion ($\sim$90%) and median delay of more than two weeks in both age groups is of particular concern and has implication for immunization planning and public health policy. Globally, about 360 million people are chronically infected with Hepatitis B virus (HBV) and can lead to serious complications such as liver cirrhosis or cancer [28]. In The Gambia, HBV infection is endemic, with 15% to 20% of the population being chronically infected [39]. HBV can be transmitted from mother-to-child during the birthing process and through breast feeding. Thus, the WHO has recommended that HepB0, one of the safest and most effective vaccines, be given within 24 hours of birth and followed by at least two subsequent doses to prevent perinatal infection [28]. The $\sim$90% delay found in this study is an improvement compared to the 98.9% HepB0 delay recorded in the Gambia in 2015 [4]. However, it highlights the need for more action to ensure timely delivery and uptake of HepB0 in the country. The evidence base suggests that the key drivers of delayed HepB0 are lack of facility delivery or mothers being discharged before their babies can be accessed for vaccination [40, 41]. While these drivers might not be under the direct purview of the EPI, the programme can work collaboratively with the relevant department of the ministry health to better align priorities.

The fact that the categorical measures of vaccination timeliness showed a contrasting pattern to the continuous measures of timeliness for most vaccines, highlights the need for studies to operationalize timeliness using the two outcome measures. The subpopulation with a longer median duration of untimely (early or late) vaccination can create windows of vulnerability, even when the overall proportion of children with early or delayed vaccination is low. For this reason, it is important for EPI programs to supplement routine measures of coverage and categorical timeliness with continuous measures of vaccination timeliness to aid a nuanced

interpretation of the quality dimension of routine immunization system performance. Our findings contextualizes the available evidence in The Gambia which shows that routine vaccination coverage of MCV1 is high [2], and the proportion of children vaccinated too early for MCV1 is generally low [5], because we provide additional evidence on the median number of days children are vaccinated outside the recommended time-frames. Efforts at reducing the median number of days that children are vaccinated too early or late, in addition to increasing timely, age-appropriate MCV1 coverage and other measures must be prioritized by the Gambia EPI to halt sporadic measles outbreaks in the country.

## Transferability and future research

To ensure comparability of data, future studies examining vaccination timeliness in other LMICs contexts can implement the approach we have adopted in defining the dimensions of vaccination timeliness. We acknowledge that vaccination windows may vary across countries, nevertheless, using the nationally accepted EPI vaccination windows, rather than an arbitrary definition makes it easier to aggregate and compare data across countries, especially LMICs with similar vaccination schedules. Nationally-representative surveys such as DHS and the multiple indicator cluster survey (MICS) are widely implemented in many LMICs and routinely generate national-level routine vaccination coverage rates. The widespread availability of these surveys presents an opportunity to replicate the analysis implemented in this study by comparing routine vaccination coverage rates with estimates of all the dimensions of vaccination timeliness. Through this approach, countries can monitor the quality dimension of their immunization systems, in addition to measuring routine vaccination coverage rates which can mask substantial immunity gaps created by untimely vaccinations.

To effectively implement targeted public health interventions, it might be necessary to move beyond utilizing national and subnational estimates of vaccination timeliness and instead identify specific subpopulations that are 'hotspots' of untimely vaccination. Future studies should deploy geospatial modelling techniques and generate maps showing the hotspots of early, delayed, and untimely interval vaccination at high-resolution, similar to spatial modelling of routine vaccination coverage already being conducted [42–44]. The factors that contribute to the pattern of untimely vaccination observed in this study are likely to be multifaceted and complex. In order to gain a deeper and more comprehensive understanding of these factors, future studies should adopt an action-oriented conceptual framework that takes into account both accessibility and utilization of immunization services, as well as demand- and supply-side factors. This approach will allow for a more robust examination of the various factors contributing to untimely vaccination, providing valuable insights for the development and implementation of effective strategies to improve vaccination timeliness.

## Methodological implications and limitations

To compute vaccination timeliness, dates of birth and vaccination are essential, and the percentage of children with vaccination cards must be high to ensure the analysis is powered to generate representative outcomes [45]. Vaccination card availability was high in our dataset and in the age group we included ($\sim 85\%$), thus, supporting the feasibility of implementing our timeliness analysis. However, in many LMICs context, the retention of vaccination cards is variable and may limit the computation of timeliness outcomes. To conduct timeliness analysis in situations where dates of birth and vaccination are incomplete, there is a need to develop, validate, and deploy methodologies that can input or predict age at vaccination especially in situations where it can be confirmed from maternal recall that the child has been vaccinated. Such imputation or prediction techniques can utilize machine learning approaches

that may leverage pre-specified characteristics such as the age at vaccination of children in similar age bands or living in the same spatial location with the index child [35, 46].

The use of DHS data for the analyses of vaccination timeliness presents certain limitations inherent to the nature of the survey data. First, the availability of valid date of birth and date of vaccination for a substantial number of children in the dataset is crucial for accurate analysis. However, the completeness and accuracy of these data elements can vary significantly across many LMIC context where the availability of home-based vaccination records are seldom incomplete. This can potentially introduce biases or limit the generalizability of the timeliness estimates. Second, cross-sectional surveys like DHS provide a snapshot of the population at a specific point in time. Since the data is typically collected every 5 years and focusses on children who were vaccinated 12–35 months before the survey was implemented, it does not reflect the most recent vaccination status and poses challenges in capturing the temporal nature of vaccination timeliness. Locally tailored approaches are needed to generate timely, high-quality, population-based vaccination data needed to assess to temporal trends in vaccination timeliness. The availability of such real-time, routinely collected data has fundamental advantages over data generated through periodic (~5 yearly) surveys like the DHS and can be exploited in the future for the analysis implemented in this study. Despite these limitations, DHS data is a valuable resource for tracking vaccination timeliness. DHS data is collected in a standardized way across countries, which makes it possible to compare vaccination rates across countries. DHS data is also collected from a large sample of households, which makes it possible to get a reliable estimate of vaccination rates.

## Supporting information

**S1 Fig. Flowchart displaying children included in this study by age group and birth/vaccination data completeness.**
(DOCX)

**S1 Table. Median number of days children were vaccinated too early and interquartile ranges for all vaccines for children 12–23 and 24–35 months in The Gambia.**
(DOCX)

**S2 Table. Median delays and interquartile ranges for all vaccines for children 12–23 and 24–35 months in The Gambia.**
(DOCX)

## Author Contributions

**Conceptualization:** Oghenebrume Wariri, Chigozie Edson Utazi, Uduak Okomo, Malick Sogur, Kris A. Murray, Chris Grundy, Sidat Fofanna, Beate Kampmann.

**Data curation:** Oghenebrume Wariri, Chigozie Edson Utazi.

**Formal analysis:** Oghenebrume Wariri, Chigozie Edson Utazi.

**Funding acquisition:** Oghenebrume Wariri, Beate Kampmann.

**Investigation:** Oghenebrume Wariri, Beate Kampmann.

**Methodology:** Oghenebrume Wariri, Chigozie Edson Utazi, Uduak Okomo, Malick Sogur, Kris A. Murray, Chris Grundy, Sidat Fofanna, Beate Kampmann.

**Project administration:** Oghenebrume Wariri.

**Resources:** Oghenebrume Wariri, Malick Sogur.

**Software:** Chigozie Edson Utazi.

**Supervision:** Chigozie Edson Utazi, Uduak Okomo, Kris A. Murray, Chris Grundy, Sidat Fofanna, Beate Kampmann.

**Validation:** Oghenebrume Wariri.

**Visualization:** Oghenebrume Wariri, Chigozie Edson Utazi.

**Writing – original draft:** Oghenebrume Wariri.

**Writing – review & editing:** Chigozie Edson Utazi, Uduak Okomo, Malick Sogur, Kris A. Murray, Chris Grundy, Sidat Fofanna, Beate Kampmann.

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
