## [Decision Letter · Decision Letter 0]

17 May 2023

PONE-D-23-05149Timeliness of routine childhood vaccination among 12-35 months old children in The Gambia: analysis of national immunisation survey data, 2019-2020PLOS ONE

Dear Dr. Wariri,

Thank you for submitting your manuscript to PLOS ONE. After careful consideration, we feel that it has merit but does not fully meet PLOS ONE’s publication criteria as it currently stands. Therefore, we invite you to submit a revised version of the manuscript that addresses the points raised during the review process.

We look forward to receiving your revised manuscript.

Kind regards,

Orvalho Augusto, MD, MPH

Academic Editor

PLOS ONE

Additional Editor Comments:

This is an important report on the analysis of timeliness to vaccination. This is an important concept almost forgotten in routine immunization programs. The authors show that despite high immunization coverage reached in the Gambia there is little timely immunization according to the national EPI immunization calendar. It should be standard to analyze timeliness as well after one reads this report.

A small minor shortcoming:

Between lines 193 and 195, please cite R. Cite also the survey package (and optionally ggplot2). Also, note that there is no svy function in that package. Please revise.

Reviewers' comments:

Reviewer's Responses to Questions

**Comments to the Author**

1. Is the manuscript technically sound, and do the data support the conclusions?

Reviewer #1: Yes

Reviewer #2: Yes

2. Has the statistical analysis been performed appropriately and rigorously? 

Reviewer #1: Yes

Reviewer #2: Yes

3. Have the authors made all data underlying the findings in their manuscript fully available?

Reviewer #1: Yes

Reviewer #2: Yes

4. Is the manuscript presented in an intelligible fashion and written in standard English?

Reviewer #1: Yes

Reviewer #2: Yes

5. Review Comments to the Author

Reviewer #1: Firstly, I commend authors’ efforts in undertaking this research project. The topic is both timely and significant, and your findings have the potential to contribute to our understanding in this area.

In general, I find the paper to be well-structured, and the methodology appears sound. You have clearly outlined the objectives of the study, and the literature review provides a comprehensive background for your work. The data analysis is robust, and the results are presented in an organized manner.

However, there are a few areas that could benefit from further improvement:

1. Detailed explanation of the method in supplementary document would help readers to interpret the findings.

2. Limitation of the methodology used in the study should be thoroughly discussed.

3. Line 294 -295 , “ To the best of our knowledge, this is the first study in an LMIC context to examine all the dimensions 295 of vaccination timeliness (timely, early, delayed, and untimely interval)”, is it true? There has been several studies conducted in LMICs (such as in Mongolia, and Nepal) focusing on timeliness in routine childhood vaccinations. Author must conduct thorough literature review and be careful while writing such statements.

4. All the figures provided are unclear due low resolution. These figures should be plotted in high resolution.

Reviewer #2: The authors have assessed the timeliness of childhood vaccination for the vaccines given in the first year of life in The Gambia using the 2019-2020 Demographic and Health Survey. They have shown that despite high coverage rates, timeliness of vaccination were low for many of the vaccines, thereby highlighting the need for the national immunisation programme and the health system to improve on this indicator.

Comments

1. How has the current (low) levels of vaccination timeliness affected the vaccine preventable disease burden in The Gambia?

2. What is the current rates of use of electronic immunisation registries (EIRs) in The Gambia? If high, wouldn't EIRs provide a better means of assessing timeliness of vaccination in comparison to DHS surveys. If EIRs use is low, highlight in the discussion the need and potential impact of EIRs in identifying children for catch-up vaccination and improved timeliness.

3. Why is the vaccination coverage referred to as crude vaccination coverage? The authors specify that survey weights have been applied to the analysis, and if so, why is this still crude vaccination coverage.

4. Line 335: For HBV vaccine, highlight the potential of new HBV maternal vaccines in the pipeline in preventing infections at birth.

5. Figure 5 and corresponding results. Split the untimely interval by vaccination before the minimum window and after the maximum window, and explain the results.

6. What proportion of children are zero-dose children and underimmunised children? For records in the DHS for zero-dose children and underimmunised children, are these included in this analysis and if so, explain how are the indefinite intervals for missed vaccines are taken into it?

7. Make the code used in the analysis publicly accessible in a online repository such as Github.

6. PLOS authors have the option to publish the peer review history of their article (what does this mean?). If published, this will include your full peer review and any attached files.

Reviewer #1: No

Reviewer #2: No

---

## [Author Response · Author response to Decision Letter 0]

10 Jun 2023

Journal Requirements and Additional Editor Comments

1. Please ensure that your manuscript meets PLOS ONE's style requirements, including those for file naming. The PLOS ONE style templates can be found HERE and HERE

RESPONSE: Thanks for providing the guidelines. We have revised the manuscript to meet the guidelines.

RESPONSE: Thank you for highlighting this error. We have revised the information included in the respective sections to ensure that they are correct and match.

3. We note that you have indicated that data from this study are available upon request. PLOS only allows data to be available upon request if there are legal or ethical restrictions on sharing data publicly. For more information on unacceptable data access restrictions, please see http://journals.plos.org/plosone/s/data-availability#loc-unacceptable-data-access-restrictions. In your revised cover letter, please address the following prompts:

b) If there are no restrictions, please upload the minimal anonymized data set necessary to replicate your study findings as either Supporting Information files or to a stable, public repository and provide us with the relevant URLs, DOIs, or accession numbers. For a list of acceptable repositories. We will update your Data Availability statement on your behalf to reflect the information you provide

RESPONSE: In our data availability section, we did not state that “data from this study are available upon request”, instead, we indicated “Yes - all data are fully available without restriction”. 

Since the data are owned by a third party (The DHS Program), and we do not have permission to share the data, . We would like to update our data availability statement to read thus “The data underlying the results presented in this study were collected as part of the Gambia Demographic and Health Survey (DHS) conducted in 2019-2020. This data is publicly available for download from the DHS Program website at https://dhsprogram.com/data/available-datasets.cfm, however, it is important to note that prior permission from the DHS program must be obtained before accessing the data” in line with PLOS requirement.

We have updated the cover letter to reflect the information given above.

RESPONSE: We have reviewed the reference and we can confirm that it is complete, correct and we have not cited any paper that have been retracted.

5. This is an important report on the analysis of timeliness to vaccination. This is an important concept almost forgotten in routine immunization programs. The authors show that despite high immunization coverage reached in the Gambia there is little timely immunization according to the national EPI immunization calendar. It should be standard to analyze timeliness as well after one reads this report.

RESPONSE: We express our gratitude to the Editor for their kind and generous comment on our work.

6. A small minor shortcoming:

Between lines 193 and 195, please cite R. Cite also the survey package (and optionally ggplot2). Also, note that there is no svy function in that package. Please revise.

RESPONSE: We have now cited R and the survey package. We have now revised the sentence and edited out “svy function”.

Reviewer #1

1. Firstly, I commend authors’ efforts in undertaking this research project. The topic is both timely and significant, and your findings have the potential to contribute to our understanding in this area.

In general, I find the paper to be well-structured, and the methodology appears sound. You have clearly outlined the objectives of the study, and the literature review provides a comprehensive background for your work. The data analysis is robust, and the results are presented in an organized manner.

However, there are a few areas that could benefit from further improvement.

RESPONSE: We thank the reviewer for the positive comments, and for the time spent reviewing our manuscript. We have provided a point-by-point response that addresses all the concerns raised.

2. Detailed explanation of the method in supplementary document would help readers to interpret the findings.

RESPONSE: We have updated the supplementary document to include additional methods that could help readers interpret the findings. However, the main methods section provides a detailed description of the methods used to derive the presented results.

3. Limitation of the methodology used in the study should be thoroughly discussed.

RESPONSE: We have expanded our discussion of the limitation of the study. See lines 397-414 on pages 12 and 13.

4. Line 294 -295 , “ To the best of our knowledge, this is the first study in an LMIC context to examine all the dimensions 295 of vaccination timeliness (timely, early, delayed, and untimely interval)”, is it true? There has been several studies conducted in LMICs (such as in Mongolia, and Nepal) focusing on timeliness in routine childhood vaccinations. Author must conduct thorough literature review and be careful while writing such statements.

RESPONSE: In lines 294-295, we state that “To the best of our knowledge, this is the first study in an LMIC context to examine all the dimensions of vaccination timeliness (timely, early, delayed, and untimely interval), AND reported the outcomes as categorical and continuous variables…”. 

We agree that there has been SEVERAL studies conducted in LMICs focusing on the timeliness of childhood vaccination as shown in our own scoping review which included 224 articles from 103 LMICs and published in 2022 (See article link here: https://journals.plos.org/globalpublichealth/article?id=10.1371/journal.pgph.0000325 and the link to the protocol which we developed and was published a priori in PLOS ONE: https://journals.plos.org/plosone/article?id=10.1371/journal.pone.0253423). Our statement was borne out of the evidence from the aforementioned published piece of work which we also clearly reference in the introduction section of this manuscript (i.e., reference 25).

Our intention in that statement was to highlight the unique contribution of our analysis, which is the concurrent reporting of all dimensions of vaccination timeliness as both categorical (proportion delayed) and continuous variables (mean/median number of days children were vaccinated too early or late). This is indicated in the second part of that statement “…and reported the outcomes as categorical and continuous variables”. We have revised the statement for clarity to accurately reflect our claims. It now reads: “To the best of our knowledge, this is the first study conducted in an LMIC context to simultaneously investigate all the dimensions of vaccination timeliness (timely, early, delayed, and untimely interval), and presents the outcomes as both categorical and continuous variables. This approach allows for a uniquely nuanced interpretation of the results unlike previous studies that often focused on one dimension or predominantly reported the outcomes as categorical variables”.

5. All the figures provided are unclear due low resolution. These figures should be plotted in high resolution.

RESPONSE: Thank you for this comment. We have submitted high quality images in the format that meets the requirement of PLOS ONE. Note that sometimes, the quality reduces after they have been rendered as a pdf file from the editorial manager.

Reviewer #2

1. The authors have assessed the timeliness of childhood vaccination for the vaccines given in the first year of life in The Gambia using the 2019-2020 Demographic and Health Survey. They have shown that despite high coverage rates, timeliness of vaccination were low for many of the vaccines, thereby highlighting the need for the national immunisation programme and the health system to improve on this indicator.

RESPONSE: We extend our sincere gratitude to the reviewer for their willingness to review our manuscript and for offering valuable feedback. In response to each comment, we have provided a point-by-point response, addressing the concerns raised.

2. How has the current (low) levels of vaccination timeliness affected the vaccine preventable disease burden in The Gambia?

RESPONSE: Thank you for this very important question and for your interest in our study. Our study was not designed to address this specific question, as we do not have the necessary data to infer causality (i.e., link low levels of timeliness to VPDs burden in The Gambia). At this point, we cannot make any bold claims about the effect of vaccination timeliness on vaccine preventable disease burden in The Gambia. We hope that we are able explore this important question in the near future by integrating population-level routine vaccination data and vaccine preventable diseases surveillance data.

We had suggested in the introduction that “This is particularly worrying as in parts of The Gambia, there has been a recent upsurge of measles, with a 6-fold increase in cases as of mid-2022 compared to 2020 figures despite a high coverage of both doses of measles-containing vaccine (MCV1 and MCV2)”. However, we cannot be certain that the increase in cases of measles are due to “low levels of vaccination timeliness”.

3. What is the current rates of use of electronic immunisation registries (EIRs) in The Gambia? If high, wouldn't EIRs provide a better means of assessing timeliness of vaccination in comparison to DHS surveys. If EIRs use is low, highlight in the discussion the need and potential impact of EIRs in identifying children for catch-up vaccination and improved timeliness.

RESPONSE: The Gambia began implementing Electronic Immunization Registers (EIRs) five years ago. This was done in phases, starting with specific administrative regions in the country. Overall, the progress in scaling up EIR has been encouraging.

There is no doubt that EIRs provides another means of assessing timeliness and its utility is well highlighted in the published literature on vaccination timeliness, especially in high-income countries. However, it is important to acknowledge a key limitation of EIRs: they only capture data of children who visit immunization facilities where EIRs have been implemented, thus, leaves out a key demographic. Therein lies the strength of DHS surveys as they can be considered representative of the overall population due to their survey design that ensure national representativeness. 

4. Why is the vaccination coverage referred to as crude vaccination coverage? The authors specify that survey weights have been applied to the analysis, and if so, why is this still crude vaccination coverage.

RESPONSE: Thank you for raising this important point. Throughout the manuscript, we have now edited out “crude” in response to this comment.

5. Line 335: For HBV vaccine, highlight the potential of new HBV maternal vaccines in the pipeline in preventing infections at birth.

RESPONSE: Thank you for raising this important point. Given that maternal hepatitis B vaccines have not yet been implemented in The Gambia and many other LMICs, we have chosen to center our discussion around the public health implications of delayed birth dose of routine childhood hepatitis B vaccine. Our intention is to maintain the focus on the main message conveyed in that section of the discussion, which highlights the risks associated with not delivering the hepatitis B birth dose in a timely manner, particularly in countries like The Gambia where hepatitis B virus infection is endemic and poses a significant public health concern as outlined by the WHO guidelines.

6. Figure 5 and corresponding results. Split the untimely interval by vaccination before the minimum window and after the maximum window, and explain the results.

RESPONSE: Done. Please see updated Figure 5 and the accompanying results section.

7. What proportion of children are zero-dose children and under-immunised children? For records in the DHS for zero-dose children and under-immunised children, are these included in this analysis and if so, explain how are the indefinite intervals for missed vaccines are taken into it? vaccination and develops recommendations that could shape the design and implementation of future research.”

RESPONSE: The focus of this analysis was specifically on assessing the timeliness of vaccination, and as such, we did not delve into the topic of zero-dose. We computed timeliness based on children who had received their vaccinations AND possessed valid dates of birth and vaccination, as these key variables were essential for calculating timeliness, as outlined in lines 144-147 of the manuscript. This approach aligns with the guidelines provided in the 2019 WHO white paper on the harmonization, collection and analysis of vaccination coverage data, which recommends the utilization of valid information from vaccination cards when computing vaccination timeliness.

Furthermore, we state in lines 186-190 “The denominator for computing vaccination timeliness (timely, early, delayed and untimely interval vaccination) was all eligible children whose date of birth was known, were within the specified age ranges, and whose vaccination dates were available from a vaccination card” which further highlights the population included in this analysis.

8. Make the code used in the analysis publicly accessible in a online repository such as Github.

RESPONSE: Thank you for your suggestion. We agree that it is important to make the codes for our research publicly available. As part of our Data Management and Sharing (DMS) plan, we will be making the codes for this particular analysis and for other sub-studies in our EDCTP-funded project publicly available in an online repository at the end of the project. We believe that making the codes publicly available will help to advance scientific research and improve the reproducibility of our findings. It will also allow other researchers to build on our work and develop new insights.

---

## [Decision Letter · Decision Letter 1]

4 Jul 2023

Timeliness of routine childhood vaccination among 12-35 months old children in The Gambia: analysis of national immunisation survey data, 2019-2020

PONE-D-23-05149R1

Dear Dr. Wariri,

We’re pleased to inform you that your manuscript has been judged scientifically suitable for publication and will be formally accepted for publication once it meets all outstanding technical requirements.

Kind regards,

Orvalho Augusto, MD, MPH

Academic Editor

PLOS ONE

Additional Editor Comments (optional):

This is a very nice manuscript. A minor suggestion (not required to do):

- In Figures 3B and 4B, is it possible to put the y-axis in log scale. So we can see the lower values where the cloud of points is concentrated

- Please, share your R code to perform this analysis

Reviewers' comments:

Reviewer's Responses to Questions

**Comments to the Author**

1. If the authors have adequately addressed your comments raised in a previous round of review and you feel that this manuscript is now acceptable for publication, you may indicate that here to bypass the “Comments to the Author” section, enter your conflict of interest statement in the “Confidential to Editor” section, and submit your "Accept" recommendation.

Reviewer #1: All comments have been addressed

Reviewer #2: All comments have been addressed

2. Is the manuscript technically sound, and do the data support the conclusions?

Reviewer #1: Yes

Reviewer #2: Yes

3. Has the statistical analysis been performed appropriately and rigorously? 

Reviewer #1: (No Response)

Reviewer #2: Yes

4. Have the authors made all data underlying the findings in their manuscript fully available?

Reviewer #1: Yes

Reviewer #2: No

5. Is the manuscript presented in an intelligible fashion and written in standard English?

Reviewer #1: Yes

Reviewer #2: Yes

6. Review Comments to the Author

Reviewer #1: (No Response)

Reviewer #2: The authors have addressed my comments well and I have no additional comments.

While you have stated that the codes for this particular analysis and for other sub-studies in our EDCTP-funded project publicly available in an online repository at the end of the project, we expect you to make the code for this study available publicly ahead of publication.

7. PLOS authors have the option to publish the peer review history of their article (what does this mean?). If published, this will include your full peer review and any attached files.

Reviewer #1: **Yes: **Santosh Kumar Rauniyar

Reviewer #2: No

---

## [Editor Report · Acceptance letter]

12 Jul 2023

PONE-D-23-05149R1 

Timeliness of routine childhood vaccination among 12-35 months old children in The Gambia: analysis of national immunisation survey data, 2019-2020 

Dear Dr. Wariri:

I'm pleased to inform you that your manuscript has been deemed suitable for publication in PLOS ONE. Congratulations! Your manuscript is now with our production department. 

Kind regards, 

on behalf of

Dr. Orvalho Augusto 

Academic Editor

PLOS ONE